# The Association of Lumbar Disc Herniation with Lumbar Volumetric Bone Mineral Density in a Cross-Sectional Chinese Study

**DOI:** 10.3390/diagnostics11060938

**Published:** 2021-05-24

**Authors:** Jian Geng, Ling Wang, Qing Li, Pengju Huang, Yandong Liu, Glen M. Blake, Wei Tian, Xiaoguang Cheng

**Affiliations:** 1School of Medical Technology, Shaanxi University of Chinese Medicine, Xianyang 712083, China; disc_doctor@163.com; 2Department of Radiology, Beijing Jishuitan Hospital, Beijing 100035, China; 1988yisheng@163.com (L.W.); docqing@pku.edu.cn (Q.L.); 1911210542@bjmu.edu.cn (P.H.); 1711210531@bjmu.edu.cn (Y.L.); 3Osteoporosis Research Unit, King’s College London, London WC2R 2LS, UK; glen.blake@kcl.ac.uk; 4Department of Spine Surgery, Beijing Jishuitan Hospital, Beijing 100035, China; tianweijst@vip.163.com

**Keywords:** lumbar intervertebral disc herniation, lumbar volumetric bone mineral density

## Abstract

Little is known about the effect of lumbar intervertebral disc herniation (LDH) on lumbar bone mineral density (BMD), and few previous studies have used quantitative computed tomography (QCT) to assess whether the staging of LDH correlates with lumbar vertebral trabecular volumetric bone mineral density (Trab.vBMD). To explore the relationship between lumbar Trab.vBMD and LDH, seven hundred and fifty-four healthy participants aged 20–60 years were enrolled in the study from an ongoing study on the degeneration of the spine and knee between June 2014 and 2017. QCT was used to measure L2–4 Trab.vBMD and lumbar spine magnetic resonance images (MRI) were performed to assess the incidence of disc herniation. After 9 exclusions, a total of 322 men and 423 women remained. The men and women were divided into younger (age 20–39 years) and older (age 40–60 years) groups and further into those without LDH, with a single LDH segment, and with ≥2 segments. Covariance analysis was used to adjust for the effects of age, BMI, waistline, and hipline on the relationship between Trab.vBMD and LDH. Forty-one younger men (25.0%) and 59 older men (37.3%) had at least one LDH segment. Amongst the women, the numbers were 46 (22.5%) and 80 (36.4%), respectively. Although there were differences in the characteristics data between men and women, the difference in Trab.vBMD between those without LDH and those with single and ≥2 segments was not statistically significant (*p* > 0.05). These results remained not statistically significant after further adjusting for covariates (*p* > 0.05). No associations between lumbar disc herniation and vertebral trabecular volumetric bone mineral density were observed in either men or women.

## 1. Introduction

Lumbar intervertebral disc herniation (LDH) refers to a displacement of the intervertebral disc tissue beyond the normal confines of the disk space, and the associated comorbidities are one of the major threats to work and functional ability [1]. Although low back pain (LBP) is multifactorial, LDH might be the most common reason [2,3], namely, nerve root impingement or irritation [4] and a local inflammatory response stimulated by the displacement of the disc material [5]. The herniated disk material may include elements of the nucleus pulposus, annulus fibrosus, cartilage, fragmented apophyseal bone, or any combination thereof [5]. There is a high prevalence of such spinal abnormalities on imaging in asymptomatic patients, and approximately 85% of patients with sciatica are found to have a herniated intervertebral disk [3,6].

Similar to LDH, osteoporosis (OP) is an age-related skeletal disease that is very prevalent in the elderly population. However, the relationship between LDH and bone health is unclear, although the relationship between the lumbar intervertebral disc and the vertebral body is both physiologically and biomechanically close. The intimate contact of the endplate with marrow is critical to nutrient diffusion into the nuclear matrix [5]. Bone mineral density (BMD) is considered to be a surrogate for bone strength and can be used to diagnose OP and to predict fractures [7]. Thus, the investigation of the associations between LDH and BMD would help us understand the impact of LDH on the incidence of osteoporotic vertebral fractures.

Dual-energy X-ray absorptiometry (DXA) and quantitative computed tomography (QCT) are recognized bone densitometry methods that provide measurements of BMD for clinical practice and research [8]. Most of the published studies on the relationship between disc lesions and BMD were measured by DXA, and therefore were unable to distinguish between the cortical and trabecular bone. In contrast, QCT provides three-dimensional trabecular volumetric BMD (vBMD) measurements that offer the opportunity to further explore the associations between LDH and vBMD. To precisely define LDH, we also used state-of-the-art MRI imaging to identify disc lesions [9]. The aims of this study were as follows: (1) to explore the relationship between lumbar disc herniation and lumbar vBMD, and (2) to compare vBMD among participants with different stages of LDH.

## 2. Materials and Methods

### 2.1. Study Subjects

The subjects were from a study on the degeneration of the spine and knee performed between June 2014 and 2017. The study was approved by the ethics committee of our hospital. The criteria for inclusion were healthy adults, aged 20–60 years, and residents in Beijing >5 years. The menopausal status could not be confirmed in all the female subjects, as some of them were not able to accurately determine when they began to be menopausal or did not remember the accurate age of menopause. The 754 participants enrolled in the study had both a QCT scan and an MRI scan of the lumbar spine. Three participants with the errors in their lumbar vBMD values (1 with an analysis error, 2 missing) and 7 participants for whom the characteristics data were missing (2 involving BMI, 2 hiplines, 1 waistline, and 1 all demographics) were excluded. Additional criteria for exclusion included autoimmune disease (e.g., rheumatoid arthritis), congenital disorders (e.g., juvenile idiopathic scoliosis), prior lumbar spine surgery, a history of metabolic bone disease or chronic diseases related to calcium absorption (hyperparathyroidism), a history of malignant tumors, the use of medications known to affect bone metabolism, and pregnancy [10,11]. The final sample size was 745. Each subject signed for informed consent.

### 2.2. Baseline Data Collection

Basic information including age (years), sex, height (cm), weight (kg), body mass index (BMI, kg/m^2^), waistline (cm), and hipline (cm) were recorded before scanning. Height was measured using a stadiometer (Harpenden stadiometer, Holtain Limited, Crosswell, UK) and weight was measured using an electronic scale (Yuanyan, Dahaibian Tech., Danyang, China). BMI was calculated as weight in kilograms divided by the square of height in meters.

### 2.3. Lumbar Vertebra Scanning by QCT

As part of the study protocol, the lumbar scan was performed on a Toshiba CT scanner (Aquilion PRIME, Toshiba, Otawara, Japan). A QCT calibration phantom (Mindways Inc., Austin, TX, USA) was placed beneath the spine and scanned simultaneously according to the standard scanning protocol by Wang et al. [12]. Scans were acquired in the supine position from the top of the 12th thoracic vertebra (T12) to the fourth sacral (S4). The spine was kept parallel to the long axis of the calibration phantom, and minimal air gaps existed between the phantom and the volunteer. The scanning parameters were as follows: 120 kV, 187 mAs, field of view 50 cm, 1 mm slice thickness, and reconstruction matrix 512 × 512. Other methodological details have been described previously [13].

### 2.4. Lumbar Vertebral Trabecular Volumetric Bone Mineral Density (Trab.vBMD) Measurement

After scanning, the CT DICOM images were transferred to the QCT workstation for further analysis with the QCT Pro 5.0.3 software (Mindways Inc.). Trab.vBMD was measured within a specific region of interest, which was defined as the oval-shaped areas containing the largest areas of the trabecular bone in the mid-plane of each vertebral body, not including the cortical bone or basivertebral vein [13,14]. The Trab.vBMD values (mg/cm^3^) of L2–4 were recorded and analyzed, respectively, and the average was calculated [14]. The osteoporosis and low bone mass were defined according to the latest Chinese expert consensus on the diagnosis of osteoporosis of vBMD > 80 mg/cm^3^ and 80 to 120 mg/cm^3^ [15].

### 2.5. Lumbar Scanning by MRI

All volunteers had lumbar MRI scans on a 1.5 T scanner (Espree, SIEMENS, Munich, Germany), and were scanned with the same multi-channel gradient waist coil to reduce the potential alterations and uncertainties on the image features [16]. T1-weighted turbo spin-echo (TSE) imaging (repetition time/echo time (TR/TE) 400/8.6 ms, slice thickness 4 mm, intersection gap 0.8 mm, voxel 200 × 264, the field of view (FOV) 180 × 280 mm) and T2-weighted TSE imaging (TR/TE 2500/100 ms, slice thickness 4 mm, intersection gap 0.8 mm, voxel 240 × 282, FOV 180 × 280 mm) were performed in the sagittal plane. In the axial plane, T2-weighted TSE imaging was performed with TR/TE 2500/100 ms, slice thickness 4 mm, intersection gap 0.4 mm, voxel 248 × 198, and FOV 160 × 180 mm.

### 2.6. Definition of Lumbar Intervertebral Disc Herniation

The MRI scans were assessed by two readers who were not provided with any clinical information and not involved in the selection or care of the participants. To standardize the nomenclature, each reader was given a manual containing definitions of imaging characteristics [4]. Following the latest review for disc herniation terminologies to enhance interobserver reliability [17] and improve contrast, we defined the non-LHD group as follows: normal/bulge (regardless of asymmetric or symmetrical [18]) and protrusion (regardless of whether focal- or broad-based protrusion [18]) with no evident contact of disk material with the nerve root [19], and without severe dural sac (thecal sac) compression or diminished dimensions of the neural foramen [20] (Figure 1). The LDH group was defined as follows: the bulge (presence of disc tissue “circumferentially” (50–100%) beyond the ring apophyses may be called “bulging”), protrusion (presence of disc tissue less than 25% of the disc circumference) with the visible contract of disk material with the nerve root, severe thecal sac compression, and diminished dimensions of the neural foramen, extraction, and sequestration (Figure 2 and Figure 3). In the LDH group, to explore the impact of the numbers of herniated segments, we defined a single-level disc herniation as subgroup 1 (Figure 2), and ≥2 segments as subgroup 2 (Figure 3). According to the age distribution of disc herniation in the data, we stratified the data by age, calling those aged 20 to 39 years the younger group and those aged 40 to 60 years the older group. Any divergence was resolved by consensus of the readers.

### 2.7. Statistical Analysis

The data were stratified into the male and female groups and then the Shapiro–Wilk test was used to test normality in the continuous variables. In both sexes the height and lumbar vertebral vBMD were normally distributed. To compare the characteristics and lumbar vBMD difference between genders, an independent samples t-test was performed on data that met the normality requirement and the Mann–Whitney U test was performed on data that were not normally distributed. Then, the above variates were stratified by lumbar intervertebral disc condition, including positive herniation or non-herniation. A repeated Shapiro–Wilk test was used to test normality in every group, respectively. Repeated independent samples t-tests and Mann–Whitney U tests were performed for the appropriate variables, respectively. Then one-way variance analysis (ANOVA) was used to find vBMD and height differences between group 1, subgroup 1, and subgroup 2. Kruskal–Wallis H test was used to find BMI, waistline, hipline differences between group 1, subgroup 1, subgroup 2 in men and women, respectively. Finally, the ANOVA analysis was performed a second time to adjust for the effect of age, BMI, waistline, and hipline on lumbar vBMD between lumbar disc herniation status. A *p* value <0.05 was considered statistically significant. SPSS 26.0 software was used to perform statistical analysis.

## 3. Results

### 3.1. Characteristics of Subjects

Seven hundred and fifty-four participants were enrolled in the study. After the nine exclusions noted above, the data from the remaining 745 volunteers (322 men, median age 39 years old; 423 women, median age 40 years old) were included in the analysis. The basic information and mean lumbar vertebral body Trab.vBMD of all the subjects stratified by sex are shown in Table 1. The differences in characteristic data and mean lumbar Trab.vBMD between the genders were observed, except for age (*p* = 0.558). Height (172.2 ± 5.9 vs. 160.5± 5.6 cm, *p* < 0.001), weight (77 vs. 59.5 kg, *p* < 0.001), BMI (25.9 vs. 23.2 kg/m^2^, *p* < 0.001), waistline (90 vs. 79 cm, *p* < 0.001), and hipline (101 vs. 95 cm, *p* < 0.001) for men were higher than for women, but women had higher lumbar vertebral vBMD compared to men (163.8 ± 35.4 vs. 148.0 ± 31.1, *p* < 0.001).

### 3.2. Stratification Analysis Relationship of LDH with Characteristics Data and Lumbar Trab.vBMD

Forty-one men in the younger group (25.0%) and 59 in the older group (37.3%) had one or more LDH segments. Nine men in the younger group (5.5%) and 24 in the older group (15.2%) presented with ≥2 herniated segments. Amongst the women, 46 in the younger group (22.5%) and 80 in the older group (36.4%) had one or more LHD segments, while 8 in the younger group (3.9%) and 31 in the older group (14.1%) presented with ≥2 herniated segments. There were no significant differences between the percentages of men and women with LDH in the same age group.

Stratification and comparative analysis were performed in both the male and female groups, according to whether there was disc herniation or not (Table 1). After age and sex stratification, data were grouped and analyzed according to the stages of lumbar disc herniation, namely, the number of herniations. The results for the men and women are shown in Table 2 and Table 3, respectively. In the younger men, the weight and hipline of the participants in the ≥2 segment herniation group were statistically significantly greater than the non-LDH participants. The participants with ≥2 segment herniations had statistically significantly lower lumbar Trab.vBMD than the participants with one disc herniation. In younger women, the participants with ≥2 segment herniations had a significantly greater weight and hipline than those without (*p* < 0.05), and they had a significantly greater BMI and waistline than the non-LDH and one segment LDH group (*p* < 0.05). However, there were no significant differences in age, height, and lumbar Trab.vBMD between the groups. In the older men, none of the differences between the groups were statistically significant. In the older women, the participants with ≥2 herniated levels had a significantly greater age, BMI, waistline, hipline than the participants of the non-LDH group (*p* < 0.05), but there were no significant differences in height, weight, and BMD between the groups (*p* > 0.05).

### 3.3. Stratification Analysis after Covariate Adjusting

To adjust for the effect of covariates on the results, we conducted two covariance analyses using age, BMI, waistline, and hipline as covariates, and disc herniation status as the grouping variable. The results of the first covariance analysis showed that age was the main covariant for both men (*p* < 0.001) and women (*p* < 0.001), and that BMI (*p* < 0.001), waistline (*p* = 0.005), and hipline (*p* = 0.014) were additional covariates for the women (Table 1). However, all the *p*-values of the intergroups in the different disc statuses are >0.05 both in the men and women (Table 2 and Table 3). This suggests that for both men and women, disc herniation status is not a significant factor in the difference in lumbar vertebral Trab.vBMD. The post-hoc tests show that there was no significant difference in covariate-adjusted BMD between the groups with and without lumbar disc herniation. The results of the second covariance analysis are shown in Table 2 and Table 3. The covariance analysis of age and gender stratified data based on the severity of the disc herniation showed that BMI and waistline in younger men, BMI and hipline in younger women, age in older men and age, BMI and waistline in older women were the main covariates. The post-hoc analysis is shown in Table 4. In the younger male group, the adjusted Trab.vBMD in non-LDH, 1-level LDH, and ≥2-levels LDH was 160.8mg/cm^3^, 168.3 mg/cm^3^, and 145.3 mg/cm^3^, respectively. In the older male group the corresponding values were 134.1 mg/cm^3^, 140.9 mg/cm^3^ and 124.6 mg/cm^3^, respectively. In the younger female group the values were 177.6 mg/cm^3^, 182.5 mg/cm^3^, and 171.3 mg/cm^3^, respectively, and in the older female group they were 150.7 mg/cm^3^, mg/cm^3^, 150.3 mg/cm^3^, respectively. All the *p*-values of the intergroup paired comparison were >0.05. After covariate adjustment, no statistically significant association between disc herniation and lumbar vertebral vBMD was found, even when lumbar BMD differences were found in the direct analysis.

## 4. Discussion

Several earlier studies have appraised the relationship between intervertebral disc disease and lumbar and/or multipoint BMD [21,22,23,24]. However, they did not focus on the association between LDH and lumbar BMD in both men and women. Our study investigated the relationship between lumbar vertebral Trab.vBMD and LDH, and also indicated that there was no statistically significant correlation between LDH and lumbar Trab.vBMD.

Previous studies have investigated lumbar vertebral BMD in patients with spinal disorders such as scoliosis [21,22], intervertebral disc degeneration [22] and osteoarthritis [23,24], and theories had been introduced to explain why the pathogenesis of LDH might be related to lumbar intervertebral disc degeneration (LDD) [25,26,27,28]. Although disc herniation is most commonly caused by mechanical injury and the consequent rupture of the fibrous annulus, some extent of initial degeneration is necessary to allow the pulpous nucleus to herniate through the fibrous bands of the annulus into the vertebral canal [29]. With age, the decrease in proteoglycans content in the disc nucleus leads to a decrease in disc height due to dehydration, which in turn transfers the load to the posterior annulus, leading to an annulus tear and the presence of radial annular fissures. Besides, a large middle disc height, relative to around the disc, was often associated with increased compression of the vertebra in osteoporosis patients [30]. This can significantly compromise the biomechanical integrity of the motion segment. Similarly, annulus tears may reduce the hydrostatic pressure of the nucleus and increase the axial load on the annulus [31]. Therefore, some scholars regard disc degeneration and disc herniation as a mutual cause and effect relationship. Although LDH is one of the main degenerative diseases of the lumbar spine, this lesion is not the same as LDD.

We evaluated LDH by MRI and compared the lumbar Trab.vBMD of the non-LDH group and the LDH group. MRI, with its superior soft-tissue contrast resolution, provides excellent anatomic detail of spinal tissues and is the gold standard for evaluating the relationship between disc material and soft tissue and neural structures [32]. In 1994, Janssen and colleagues [33] documented that MR imaging had superior sensitivity and specificity compared to CT and CT-myelography. Some scholars have even suggested that the accuracy of magnetic resonance imaging in the diagnosis of disc herniation approaches 100% [34].

We performed a subgroup analysis based on the number of herniated discs. After controlling for covariates, we found that the association between LDH and lumbar Trab.vBMD was not statistically significant. Our outcomes indicated that LDH might not impact on the changes of lumbar vertebral Trab.vBMD. However, the observation needs further prospective investigation or larger sample size studies to confirm.

In addition to bulge, protrusion, extrusion, and sequestered, Schmorl’s node (SN) is also a type of disc herniation that derives from the weakening of the vertebral endplates. Although the relationship between classical horizontal disc herniation and vertebral BMD is currently unknown, a study [35] on SN and thoracolumbar Trab.vBMD has been conducted. Similar to our conclusion, their results show that the mean Trab.vBMD in the patients was 131.6 g/cm^3^ compared with 140.7 g/cm^3^ in the control group (*p* = 0.03). This may indicate that disc herniation does not cause the increased cancellous density in the vertebra.

One of the strengths of this study was the use of QCT to measure Trab.vBMD, since trabecular bone is affected earlier and to a greater degree than cortical bone. In contrast, several previous studies measured BMD using DXA. At present, it is widely believed that the degree of disc degeneration is positively correlated with BMD [36,37]. Disc degeneration results in a shift of load from the nucleus to the annulus, leading to reduced density in the trabecular core and increased density in cortical bone [38]. DXA cannot distinguish between cortical bone and trabecular bone, and it is sensitive to the presence of aortic calcifications, vertebral fracture, spine degeneration such as vertebral osteophytes, end-plate sclerosis, facet joint degeneration, and ligament calcification, which can all lead to falsely increased lumbar vertebral areal BMD (aBMD) [39]. Although DXA is a well-established technique, QCT is a more accurate method for BMD assessment and a more sensitive modality for the evaluation of BMD changes [7,40]. Xu et al. found that the detection rate of osteoporosis in men older than 60 years was significantly higher by QCT than by DXA [7]. Where aBMD measured by DXA includes both cortical and trabecular bone, vBMD derived from QCT is focused on a volumetric measure of vertebral trabecular bone that has a high turnover rate compared to cortical bone [40].

Our results found a slightly higher incidence in younger men (age range 20 to 39 years), in whom 25.0% had one or more LDH segments, and women of the same age range in whom the incidence was 22.5%. Previous studies [41,42,43] have found that young men have a higher incidence of LDH than women, which is attributed to the fact that young men engage in more physical activity than women. However, in the present study the difference was not statistically significant (*p* > 0.05), when calculated using Wilson’s test for two independent proportions. Amongst the participants aged 40 to 60 years, the incidence of LDH was 37.3% in the men compared with 36.4% in the women (*p* > 0.05).

Lumbar intervertebral disc herniation is associated with genetic and environmental factors and has a considerable negative effect on the economy and society [2,3]. Our study indicates that multiple characteristics and factors affected the lumbar intervertebral disc health status, especially in women. In particular, women of childbearing age are likely to be affected by the increased spinal load and change in curvature during pregnancy [44], and perimenopausal women by the decrease in estrogen [45].

Although it has been reported that the severity of symptoms has no relationship with the herniation size or shape [5], there has also been an animal experiment that found that herniated disc compression into the epidural space, without nerve compression, induced nerve dysfunction and nerve fiber degeneration [46]. Radiologically, disc herniation is often asymptomatic, but the morphology and composition can change. Also, many patients with low back pain often have no associated medical imaging findings. An investigation of the correlation between the clinical presentation of low back pain and MRI findings concluded that in 139 out of 354 patients (39%), the current episode of LBP was unrelated to any of the abnormalities found on the MRI [47]. This prompted us to focus the present study solely on the imaging findings of disc herniation, regardless of the clinical finding, and to investigate the relationship between lumbar disc herniation verified by MRI and lumbar Trab.vBMD.

Our study has several limitations. First, we performed a cross-sectional study, which is less informative than a longitudinal study and limits the ability to reflect differences in the relationship with age between the development of disc degeneration and reduction in bone mineral density. Second, our study population was drawn from community-dwelling adults and might underestimate the prevalence of lumbar disc herniation in the general population, because it did not include disabled or functionally limited elderly individuals or those with marked symptoms of disc herniation. Third, the participants were living in the Beijing urban area, and their lumbar vBMD measurements might not be representative of other Chinese cities and are probably poorer than might be found in rural areas. Fourthly, our inclusion criteria did not include information on the patients’ low back pain and instead our study focused on the impact of LDH on lumbar vertebral Trab.vBMD. Fifthly, the automatic detection of intervertebral disc herniation would be very helpful in reducing the intraobserver variability and would improve the detection of intervertebral disc herniation [48,49], but we did not perform such automatic strategies in our study.

In conclusion, we found no association between LDH and lumbar vertebral Trab.vBMD in either men or women from 20 to 60 years old. None of the differences between LDH and lumber Trab.vBMD in men and women were statistically significant. Our results indicate that BMD evaluation in LDH patients may not be necessary in clinical practice.

## Figures and Tables

**Figure 1 diagnostics-11-00938-f001:**
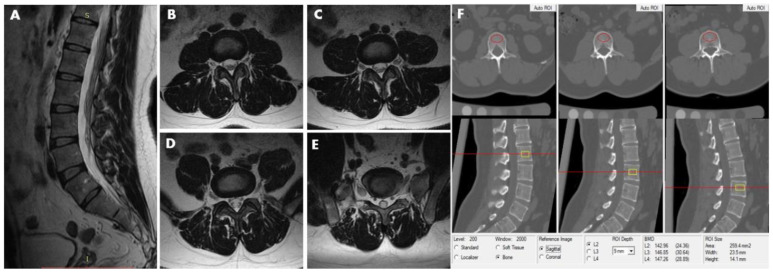
This 40-year-old male participant was assigned to the non-LDH group. MRI (T2-weighted image) revealed no lumbar disc herniation from L2–3 to L5–S1disc. (**A**) Sagittal image. (**B**) L2/3 axial image. (**C**) L3/4 axial image. (**D**) L4/5 axial image. (**E**) L5/S1 axial image. (**F**) The measurements of L2, L3, and L4 vertebral trabecular volumetric bone mineral density (Trab.vBMD) are shown; the BMD of L2, L3, and L4 is 142.96 mg/cm^3^, 146.85 mg/cm^3^ and 147.26 mg/cm^3^, respectively. The average lumbar Trab.vBMD is 145.69 mg/cm^3^.

**Figure 2 diagnostics-11-00938-f002:**
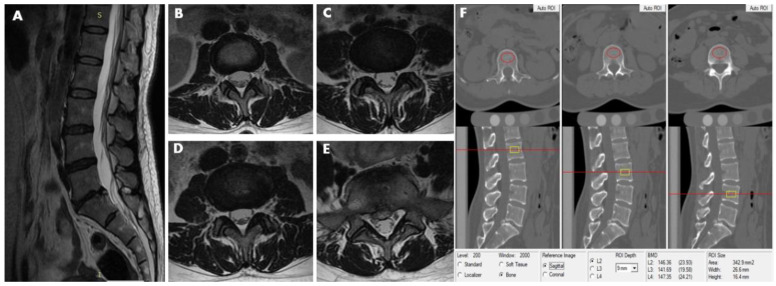
This 40-year-old female participant was assigned to the LDH group and subgroup 1. MRI (T2-weighted image) revealed lumbar disc herniation condition from L2–3 to L5–S1disc. (**A**) Sagittal image revealed lumbar disk herniation in the level of L5–S1. (**B**) L2–3 axial image shows no abnormality in disc morphology. (**C**) L3–4 axial image showns no abnormality in disc morphology. (**D**) Also, the L4–5 axial image shows no abnormality in disc morphology. (**E**) L5–S1 axial image show focal protrusion at the central canal zone with a high-intensity zone (HIZ). (**F**) The measurements of L2, L3, and L4 vertebral trabecular volumetric bone mineral density (Trab.vBMD) are shown; the BMD of L2, L3, and L4 is 146.36 mg/cm^3^, 141.69 mg/cm^3^ and 147.35 mg/cm^3^, respectively. The average lumbar Trab.vBMD is 145.13 mg/cm^3^.

**Figure 3 diagnostics-11-00938-f003:**
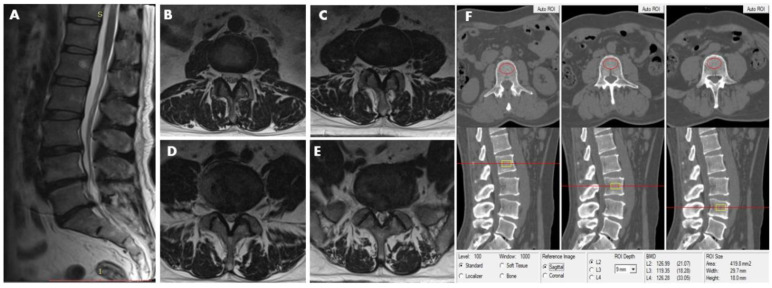
This 48-year-old male participant was assigned to the LDH group and subgroup 2, simultaneously. MRI (T2-weighted image) revealed lumbar disc herniation condition from L2–3 to L5—S1disc. (**A**) Sagittal image revealed lumbar disk herniation in L3–4, L4–5, and L5–S1. (**B**) L2–3 axial image shows no abnormality in disc morphology. (**C**) L3–4 axial image shows focal protrusion at the central canal zone with a high-intensity zone (HIZ). (**D**) L4–5 axial image shows broad-based protrusion at posterolateral or dorsally, decreasing the diameter of the spinal canal and the foramen. (E) L5/S1 axial image shows asymmetric bulge companion with a focal protrusion, the right normal epidural fat layer between the two is not evident (at least the contact). (**F**) The measurements of L2, L3, and L4 vertebral trabecular volumetric bone mineral density (Trab.vBMD) are shown; the BMD of L2, L3, and L4 is 126.99 mg/cm^3^, 119.35 mg/cm^3^, and 126.28 mg/cm^3^, respectively; the average lumbar Trab.vBMD is 124.21 mg/cm^3^.

**Table 1 diagnostics-11-00938-t001:** Characteristics of the study group.

Men						Women					
	**Mean ± SD or Median Value**					**Mean ± SD or Median Value**					
Parameters	total	Non-LDH	LDH	P1	P2	total	Non-LDH	LDH	P1	P2	P3
Sample size	322	222 (68.9%)	100 (31.1%)			423	297 (70.2%)	126 (29.8%)			
Age (years)	39	39	41	0.02	<0.001	40	38	43	<0.001	<0.001	0.558
Height (cm)	172.2 ± 5.9	172.0 ± 6.2	172.8 ± 5.3	0.265		160.5 ± 5.6	160.4 ± 5.7	160.7 ± 5.2	0.59		<0.001
Weight (kg)	77	76	78	0.017		59.5	59	61	0.002		<0.001
BMI (kg/m^2^)	25.9	25.6	26.4	0.027	0.093	23.2	22.9	24.0	0.003	<0.001	<0.001
Waistline (cm)	90	90	92	0.034	0.145	79	77.5	81	0.000	0.005	<0.001
Hipline (cm)	101	100	101.25	0.015	0.235	95	95	97	0.008	0.014	<0.001
Lumbar BMD (mg/cm^3^)	148.0 ± 31.1	149.6 ± 31.3	144.6 ± 30.6	0.188		163.8 ± 35.4	166.1 ± 33.7	158.5 ± 38.6	0.059		<0.001
Adjusted vBMD (mg/cm^3^)		148.0 (144.4 to 151.6)	148.0 (142.6 to 153.4)		0.997		163.8 (160.4 to 167.2)	163.9 (158.5 to 169.2)		0.986	

Abbreviations: BMI = body mass index, vBMD = volumetric bone mineral density. P1: independent samples t-test or Kruskal–Wallis H test between the non-LDH group and LDH group. P2: one-way covariance analysis in lumbar vBMD difference between non-LDH group and LDH group. P3: independent samples t-test in characteristic data between the male group and female group.

**Table 2 diagnostics-11-00938-t002:** Comparison of characteristic and vBMD data between different disc conditions in younger and older men.

Age Group 1 (20–39)					*p* _0_	*p*_1_ ≤ 0.05 for within Group	*p*_2_ for Covariance Analysis	Age Group 2 (40–60)				*p* _0_	*p*_1_ ≤ 0.05 for within Group	*p*_2_ for Covariance Analysis
	Non-LDH	1 Segment	≥2 Segments	Total				Non-LDH	1 Segment	≥2 Segments	Total			
Sample size Rate (%)	123 38.2%	32 9.9%	9 2.8%	164 50.9%				99 30.7%	35 10.9%	24 7.5%	158 49.1%			
Age	33	31.5	32	33	0.638 ^†^	-	0.216	45	45	48	45	0.309	-	<0.001
Height (cm)	172.0 ± 6.5	174.4 ± 5.1	174.1 ± 4.8	172.6 ± 6.2	0.114 ^‡^	-	-	171.9 ± 5.8	171.7 ± 5.0	171.6 ± 5.8	171.8± 5.6	0.961	-	-
Weight (kg)	75	79	81	77	0.017 ^†^	0.005 ^b^	^-^	78	75	76.5	77	0.753	-	-
BMI (kg/m^2^)	25.2	26.6	26.2	25.5	0.078 ^†^	-	0.004	26.0	25.6	27.0	26.0	0.546	-	0.473
Waistline (cm)	89	91.5	88	89.75	0.088 ^†^	-	0.038	91	92	92.5	92	0.770	-	0.785
Hipline (cm)	100	101	103	101	0.031 ^†^	0.001 ^b^	0.135	100	101	100.5	100.5	0.399	-	0.523
L vBMD (mg/cm^3^)	161.2 ± 26.2	167.0 ± 27.3	143.5 ± 12.1	161.4 ± 26.2	0.057 ^‡^	0.045 ^c^	-	135.1 ± 31.2	140.1 ± 28.2	121.8 ± 23.1	134.2 ± 29.8	0.061	-	-
Herniation segments							0.052							0.09

*p*_0_: Data comparison between non-LDH, 1-level LDH, and ≥2 levels group; ^†^ by Kruskal–Wallis H test, ^‡^ By one-way variance analysis (ANOVA). *p*_1_: only comparisons with values of *p*_1_ < 0.05 are shown. All comparisons between the non-LDH group and 1-level LDH group had *p*_1_ > 0.05; *p*_1_ ^b^: comparison between non-LDH group and ≥2-levels LDH; *p*_1_ ^c^: comparison between 1-level LDH group and ≥2-levels LDH. *p*_2_: to adjust for the effect of age, BMI, waistline, and hipline on lumbar vBMD between lumbar disc herniation status.

**Table 3 diagnostics-11-00938-t003:** Comparison of characteristic and Trab.vBMD data between different disc condition in younger and older women.

Age Group 1 (20–39)					*p* _0_	*p*_1_ ≤ 0.05 for within Group	*p*_2_ for Covariance Analysis	Age Group 2 (40–60)				*p* _0_	*p*_1 ≤_ 0.05 for within Group	*p*_2_ for Covariance Analysis
	Non-LDH	1-Level	≥2-Levels	Total				Non-LDH	1-Level	≥2-Levels	Total			
Sample size Rate (%)	158 37.4%	38 9.0%	8 1.9%	204 48.3%				139 32.9%	49 11.6%	31 7.3%	220 51.7%			
Age	32	32	30.5	32	0.675 ^†^	-	0.293	45	46	50	46	0.027 ^†^	0.007 ^b^	<0.001
Height (cm)	161.0 ± 5.8	161.9 ± 5.0	161.1 ± 3.0	161.2 ± 5.6	0.654 ^‡^	-		159.7 ± 5.6	160.2 ± 5.7	159.8 ± 5.1	159.8 ± 5.5	0.825 ^‡^	-	
Weight (kg)	58	59	69	58	0.018 ^†^	0.004 ^b^		60	61	64	60.5	0.053 ^†^	-	
BMI (kg/m^2^)	22.6	22.3	27.1	22.6	0.040 ^†^	0.015 ^b^ 0.013 ^c^	<0.001	23.4	24.3	25.0	23.8	0.036 ^†^	0.031 ^b^	0.012
Waistline (cm)	76	77.5	84.5	76	0.018 ^†^	0.006 ^b^ 0.028 ^c^	0.069	80	83	84	81	0.041 ^†^	0.030 ^b^	0.020
Hipline (cm)	95	94	101.5	95	0.211 ^†^	0.010 ^b^	0.001	96	98	99	97	0.039 ^†^	0.042 ^b^	0.876
BMD (mg/cm^3^)	177.9 ± 28.3	180.8 ± 34.5	172.9 ± 23.2	178.3 ± 29.3	0.753 ^‡^	-		152.6 ± 34.3	148.8 ± 37.5	143.0 ± 35.8	150.4 ± 35.3	0.369 ^‡^	0.359	
Herniation segments							0.496							0.965

*p*_0_: Data comparison between non-LDH, 1-level LDH, and ≥2 levels group; ^†^ by Kruskal–Wallis H test, ^‡^ By one-way variance analysis (ANOVA). *p*_1_: only comparisons with values of *p*_1_ < 0.05 are shown. All comparisons between the non-LDH group and 1-level LDH group had *p*_1_ > 0.05; *p*_1_
^b^: comparison between non-LDH group and ≥2-levels LDH; *p*_1_
^c^: comparison between 1-level LDH group and ≥2-levels LDH. *p*_2_: after adjustment for the effect of age, BMI, waistline, and hipline on lumbar vBMD between lumbar disc herniation status.

**Table 4 diagnostics-11-00938-t004:** Adjusted lumbar Trab.vBMD measurements and pair comparison within groups.

Gender	Age Group	LDH Levels	Adjusted Mean Value (SD)	95% Confidence Interval		Paired Comparison, *p* > 0.05
				Lower	Upper	
				Limit	Limit	
Men	1	0	160.8 (2.3)	156.2	165.2	≥0.05 (0.052–0.433) *
		1	168.3 (4.6)	159.3	177.4	
		≥2	145.3 (8.4)	128.7	162.0	
	2	0	134.1 (2.8)	128.6	139.6	≥0.05 (0.086–0.649) *
		1	140.9 (4.7)	131.6	150.2	
		≥2	124.6 (5.7)	113.3	135.8	
Women	1	0	177.6 (2.3)	173.1	182.0	≥0.05 (0.961–1) *
		1	182.5 (4.6)	173.4	191.7	
		≥2	171.3 (10.3)	150.9	191.6	
	2	0	150.7 (2.5)	145.8	155.7	≥0.05 (1.0) *
		1	149.4 (4.2)	141.1	157.7	
		≥2	150.3 (5.4)	139.8	160.9	

*: *p* values of paired comparison between 0 and 1 level of herniation, 0 and ≥2 levels of herniation, and 1 and ≥2 levels of herniation are all ≥0.05.

## Data Availability

The data presented in this study are available on request from the corresponding author.

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
