# Peer review of "The Association of Lumbar Disc Herniation with Lumbar Volumetric Bone Mineral Density in a Cross-Sectional Chinese Study"

_diagnostics, 2021, doi:10.3390/diagnostics11060938_

Round 1

Reviewer 1 Report

General comment:

This works deals with the investigation of lumber invetrertebral disc herniation (LDH) and lumbar vertebral trabecular volumetric bone mineral density (Trab.vBMD). The authors have carried out the study on 74 subjects, monitoring them for a three years period. The study reports a non-statistically significant association for lumbar disc herniation and vertebral trabecular volumetric bone mineral density. 

Specific comment throughout the paper: 

Abstract

Line 4: LDH is not defined therein (only in Line 7). Please, avoid using abbreviations and acronyms in the abstract. Define them instead. 

1. Introduction 

Line 24-28: There is a problem with the font and size of the text. Please check with the authors submission guidelines. Also Line 39 suffer from the same problem.

Line 28: Missing space between text and reference [1]. Please fix. 
This type of error is repeated elsewhere (Line 29, Line 30, Line 32, Line 34, Line 39, Line 42). 
I stongly invite the authors to revise thoroughly the manuscript and proofread it to adjust these errors. 

Lines 45-49: Missing reference for the RI detection of disc lesion. Maybe:

Kim, Ju-Hwi, et al. "Traumatic lumbar disc herniation mimicking epidural hematoma: A case report and literature review." Medicine 98.18 (2019).

would fit for your work. 

2. Materials and methods

2.1

Please provide a more deatailed comparison of your inclusion criteria with similar works from the literature. Or further explain the exlcusion criteria by supporting them with appropriate references, such as 

Simpson, Andrew K., et al. "Quantifying the effects of age, gender, degeneration, and adjacent level degeneration on cervical spine range of motion using multivariate analyses." Spine 33.2 (2008): 183-186.

Ravindra, Vijay M., et al. "Degenerative lumbar spine disease: estimating global incidence and worldwide volume." Global spine journal 8.8 (2018): 784-794.

2.2

Missing details (producer, sensititivy, accuracy) of the staidomeeter and the electronic scale.

2.4 Please check the title (Line 74) and correct the font and size.

2.5

I miss some reference for the choices of the sequences. Furthermore, I suggest the author to report the type and model of the coils used for the MRI exam. These info are relevant. For instance, given tha the 3 year period is quite long, it is also possible that different coils were used on the same subject, thus altering the signal to noise ratio and sensitivity, maybe ruining your analysis. 
Most authors, especially those very focused on the clinical aspects, sometimes forget that instrumentation-related bias can be very relevant. About this point, please read:

Dachena, Chiara, et al. "Combined Use of MRI, fMRIand Cognitive Data for Alzheimer’s Disease: Preliminary Results." Applied Sciences 9.15 (2019): 3156.

As pointed out by Dachena et al, using the same instrumentatio could significantly reduce the potential alterations and uncertainties on the image features to be analyzed. Do you have accounted for this aspect in your study?

2.6

Do not use red color and italic for Fig 1. Use also "Fig. 1" as way to address the figure in the text. 

Line 101: missing space before "bulging"

Line 104: bold font, why? Please fix. 

Figure 1. The caption has the problem of presenting different font sizes. Please fix.

About Sect. 2.6, given the high number of patient, and given the clear definition, why automatic detection was not performed? See

Ghosh, Subarna, et al. "Composite features for automatic diagnosis of intervertebral disc herniation from lumbar MRI." 2011 Annual International Conference of the IEEE Engineering in Medicine and Biology Society. IEEE, 2011.

Raja'S, Alomari, et al. "Automatic diagnosis of lumbar disc herniation with shape and appearance features from MRI." Medical Imaging 2010: Computer-Aided Diagnosis. Vol. 7624. International Society for Optics and Photonics, 2010.

Ghosh, Subarna, et al. "Computer-aided diagnosis for lumbar mri using heterogeneous classifiers." 2011 IEEE International Symposium on Biomedical Imaging: From Nano to Macro. IEEE, 2011.

As pointed out for the MRI brain analysis by Dachena et al, the intraobserver variability could be reduced by the automatic processing. In [12] this point is crucial too. The authors must comment about the possible automatic strategies and the impact they could have on their work. 

Figure 2 and Figure 3 have the same problem with the caption.

2.7

A table for summarizing the test and orgnization of this section would help the reader in navigating your method and, also, the results.

3. Results

Tab. 2 and Tab. 3 and Tab. 4 have captions with wrong sizes.

3.3 

The covariate adjusting is remarkable.

Tab. 4: "Womenn" Please fix.

4. Discussion

Line 262: Missing ".".

The dicussion section is extensive and supported by the results. 

Notes:

I suggest the authors to provide a list of abbreviation and acronyms, which is included as a possibilty in the template.

Please revise elsewhere the referencing style: "text... [1]."  

Author Response

Response to Reviewer 1 Comments

The authors gratefully acknowledge the time and effort of the reviewers in providing insightful comments to improve the quality of the submitted manuscript. In the revised manuscript, we have made an effort to address all the reviewers’ concerns. In the following, we have used black font for the comments provided by the reviewers, and red font for the corresponding responses.

Point 1:  Abstract: Line 4: LDH is not defined therein (only in Line 7). Please, avoid using abbreviations and acronyms in the abstract. Define them instead. 

Response 1: Thanks. We have fixed this.

Point 2: 1. Introduction: Line 24-28: There is a problem with the font and size of the text. Please check with the authors’ submission guidelines. Also, Line 39 suffers from the same problem.

Response 2: We have solved this problem in lines 24-28 and 39, and keep the font and size required by the guidelines of submission.

Point 3: Line 28: Missing space between text and reference [1]. Please fix. This type of error is repeated elsewhere (Line 29, Line 30, Line 32, Line 34, Line 39, Line 42).  I strongly invite the authors to revise thoroughly the manuscript and proofread it to adjust these errors. 

Response 3: Thanks a lot. We have fixed this.

Point 4: Lines 45-49: Missing reference for the RI detection of disc lesion.

Response 4: Thanks for sharing the article (Kim, Ju-Hwi, et al. "Traumatic lumbar disc herniation mimicking epidural hematoma: A case report and literature review." Medicine 98.18 (2019).), and we have cited this article at the end of the introduction.

Point 5: Please provide a more detailed comparison of your inclusion criteria with similar works from the literature. Or further explain the exclusion criteria by supporting them with appropriate references, such as

Simpson, Andrew K., et al. "Quantifying the effects of age, gender, degeneration, and adjacent level degeneration on cervical spine range of motion using multivariate analyses."  Spine 33.2 (2008): 183-186.

Ravindra, Vijay M., et al. "Degenerative lumbar spine disease: estimating global incidence and worldwide volume." Global spine journal 8.8 (2018): 784-794.

Response 5:  Thanks.

We have modified the exclusion and inclusion criteria as follows:

The criteria for inclusion were healthy adults, aged 20–60 years, and residents in Beijing > 5 years. The menopausal status could not be confirmed in all the female subjects, as some of them were not able to accurately determine when they began to be menopausal or did not remember the accurate age of menopause.  The 754 participants enrolled in the study had both a QCT scan and an MRI scan of the lumbar. 3 participants with the error in the lumbar vBMD values (1 with an analysis error, 2 missing) and 7 participants for whom the characteristics data were missing (2 involving BMI, 2 hiplines, 1 waistline, and 1 all demographics) were excluded. Additional criteria for exclusion criteria included autoimmune disease (eg, rheumatoid arthritis), congenital disorders (eg, juvenile idiopathic scoliosis), prior lumbar spine surgery, a history of metabolic bone disease or chronic diseases related to calcium absorption (hyperparathyroidism), a history of malignant tumors, the use of medications known to affect bone metabolism, and pregnancy [13,14]. The final sample size was 745.

Point 6: Missing details (producer, sensitivity, accuracy) of the stadiometer and the electronic scale.

Response 6:  Many thanks for pointing this out. We used the stadiometer and the electronic scale in clinical practice and the engineers in our hospital’s equipment department would adjust these two equipments on schedule. Unfortunately, we have no details of sensitivity, accuracy of the stadiometer and the electronic scale, but we have added the producer information.

Point 7: Please check the title (Line 74) and correct the font and size.

Response 7: We have checked and corrected this error.

Point 8: I miss some references for the choices of the sequences. Furthermore, I suggest the author to report the type and model of the coils used for the MRI exam. These info are relevant. For instance, given that the 3 year period is quite long, it is also possible that different coils were used on the same subject, thus altering the signal to noise ratio and sensitivity, maybe ruining your analysis. 
Most authors, especially those very focused on the clinical aspects, sometimes forget that instrumentation-related bias can be very relevant. About this point, please read:

Dachena, Chiara, et al. "Combined Use of MRI, fMRI and Cognitive Data for Alzheimer’s Disease: Preliminary Results." Applied Sciences 9.15 (2019): 3156.

As pointed out by Dachena et al, using the same instrumentatio could significantly reduce the potential alterations and uncertainties on the image features to be analyzed. Do you have accounted for this aspect in your study?

Response 8:  Thanks very much for this interesting comment. We used the same machine and the same coil to scan the lumbar spine during the three years. Our lumbar scan is performed using a machine that examines the bed with its own multi-channel gradient lumbar coil.  In our hospital, on Saturdays and Sundays our MRI machines are not open to patients, only for scientific research. About 10 volunteers were scanned every weekend. The volunteers also had QCT scanns (cervical and lumbar vertebrae), and plain radiographs of the knee, although part of the data was not mentioned or used in this study.

Point 9: Do not use red color and italic for Fig 1. Use also "Fig. 1" as way to address the figure in the text. 

Response 9: We have solved this problem.

Point 10: Line 101: missing space before "bulging"

Response 10: We have fixed this error.

Point 11: Line 104: bold font, why? Please fix. 

Response 11: We have fixed it.

Point 12: Figure 1. The caption has the problem of presenting different font sizes. Please fix.

Response 12: We have fixed it.

Point 13: About Sect. 2.6, given the high number of patient, and given the clear definition, why automatic detection was not performed? See

Ghosh, Subarna, et al. "Composite features for automatic diagnosis of intervertebral disc herniation from lumbar MRI." 2011 Annual International Conference of the IEEE Engineering in Medicine and Biology Society. IEEE, 2011.

Raja'S, Alomari, et al. "Automatic diagnosis of lumbar disc herniation with shape and appearance features from MRI." Medical Imaging 2010: Computer-Aided Diagnosis. Vol. 7624. International Society for Optics and Photonics, 2010.

Ghosh, Subarna, et al. "Computer-aided diagnosis for lumbar mri using heterogeneous classifiers." 2011 IEEE International Symposium on Biomedical Imaging: From Nano to Macro. IEEE, 2011.

As pointed out for the MRI brain analysis by Dachena et al, the intraobserver variability could be reduced by the automatic processing. In [12] this point is crucial too. The authors must comment about the possible automatic strategies and the impact they could have on their work. 

Response 13: Thanks a lot for the valuable comment. Truly the automatic detection of intervertebral disc herniation would be very helpful in the clinical practice and research. However, it needs an appropriate software, or the cooperation with the engineering in medicine group to develop such module. At present, unfortunately, we could not perform such a CAD study. We have added some discussions about the automatic strategies in the limitation part.

Point 14: A table for summarizing the test and organization of this section would help the reader in navigating your method and, also, the results.

Response 14: Thanks for this interesting suggestion. We have added such a table as the supplement material.

Point 15: Tab. 2 and Tab. 3 and Tab. 4 have captions with wrong sizes.

Response 15: We have fixed it.

Point 16: The covariate adjusting is remarkable.

Response 16: Thank you very much for your appreciation.

Point 17: Tab. 4: "Womenn" Please fix.

Response 17: We have fixed it.

Point 18: Line 262: Missing ".".

Response 18: We have fixed it.

Point 19: The discussion section is extensive and supported by the results. 

Response 19: Thank you very much for your appreciation.

Point 20: I suggest the authors to provide a list of abbreviation and acronyms, which is included as a possibilty in the template.

Response 20: Thanks. We have provided the list of abbreviation

Point 21: Please revise elsewhere the referencing style: "text... [1]."  

Response 21:  Thanks. We have fixed this.

Reviewer 2 Report

The study focus on the association of lumbar disc herniation and bone mineral density. QCT was performed for more precise diagnose and the patients were stratified for the analysis. The topic is interesting, but there are several aspects that can be improved.

  1. The introduction part has not clearly explained why the correlation between LDH and BMD is important for improving clinical practice.
  2. The clinical problem to be solved seems to be LDH, but in the first sentence of the abstract, it seems that influence of LDH on BMD is the subject of this study. Should this be a mistake that the influence of BMD on LDH is more relevant? The introduction should make the goal clearer, although the correlation between BMD and LDH do not provide evidence for the cause-effect relationship (This is a limitation of the study that should be discussed in the discussion part). 
  3. In the method part, the quantification of BMD is too simple to be described as 'the same as former study'. How the region of interest selected, how the threshold determined etc. should be described in details.
  4. In line 266-269, p value upper than 0.05 cannot be interpreted as no difference between groups. It can be the sample size is still not big enough to find the significance.
  5. In conclusion part, it is not clear how the result contribute to relationship between disc disorders and BMD, since no correlation was found. The result would rather suggest that in clinics, BMD evaluation in LDH patients may not be necessary.

Author Response

Response to Reviewer 2 Comments

The authors gratefully acknowledge the time and effort of the reviewers in providing insightful comments to improve the quality of the submitted manuscript. In the revised manuscript, we have made an effort to address all the reviewers’ concerns. In the following, we have used black font for the comments provided by the reviewers, and red font for the corresponding responses.

Point 1: The introduction part has not clearly explained why the correlation between LDH and BMD is important for improving clinical practice.

Response 1:  Thanks. We have revise the introduction and added such a paragraph to make it more clear:

Similar to LDH, osteoporosis (OP) is an age-related skeletal disease which is very prevalent in elderly population. However, the relationship between LDD and bone health is unclear, although the relationship between the lumbar intervertebral disc and the vertebral body is both physiologically and biomechanically close. The intimate contact of the endplate with marrow is critical to nutrient diffusion into the nuclear matrix [5].  Bone mineral density (BMD) is considered to be a surrogate for bone strength and can be used to diagnose OP and to predict fractures (ref.10). Thus, the investigation of the associations of LDH and BMD would help us understand the impact of LDH on the incidence of osteoporotic vertebral fractures.

Point 2: The clinical problem to be solved seems to be LDH, but in the first sentence of the abstract, it seems that influence of LDH on BMD is the subject of this study. Should this be a mistake that the influence of BMD on LDH is more relevant? The introduction should make the goal clearer, although the correlation between BMD and LDH do not provide evidence for the cause-effect relationship (This is a limitation of the study that should be discussed in the discussion part). 

Response 2: Many thanks for your helpful comments. The main goal of our study was to explore the impact of LDH on BMD. LDH  and OP are very prevalent in elderly people. It is unclear whether LDH patients have higher vertebrae fracture risk compared to non-LDH population or not. Thus the investigation of the associations of LDH and BMD would help us have a better understanding for the impact of LDH on the incidence of osteoporotic vertebral fractures. We, however, admit that the correlation between BMD and LDH do not provide evidence for the cause-effect relationship, and a prospective study on this is warranted.

Point 3: In the method part, the quantification of BMD is too simple to be described as 'the same as former study'. How the region of interest selected, how the threshold determined etc. should be described in details.

Response 3:  Thanks. We have revised this part to make it more clear.

Point 4: In line 266-269, p value upper than 0.05 cannot be interpreted as no difference between groups. It can be the sample size is still not big enough to find the significance.

Response 4:  Many thanks for your comment. We fully agree with you, but based on analysis in this study, we did not find statistically significant difference between two groups. We have revised this paragraph and hightlighted the sample size limitation in discussion as below:

We performed a subgroup analysis based on the number of herniated discs. After controlling for covariates, we found that the association between LDH and lumbar Trab.vBMD was not statistically significant. Our outcomes indicated that LDH might not impact  on the changes of lumbar vertebral Trab.vBMD. However, the observation needs further prospective investigation or larger sample size studies to confirm.

Point 5:  In conclusion part, it is not clear how the result contribute to relationship between disc disorders and BMD, since no correlation was found. The result would rather suggest that in clinics, BMD evaluation in LDH patients may not be necessary.

Response 5: Thanks so much for your valuable comment.

We have revised the conclusion as follows:

In conclusion, we found no association between LDH and lumbar vertebral Trab.vBMD in either men or women from 20 to 60 years old. None of the differences between LDH and lumber Trab.vBMD in men and women were statistically significant. Our results indicate that BMD evaluation in LDH patients may not be necessary in clinical practice.

Round 2

Reviewer 1 Report

Dear Editor,

thank you for having allowed me to serve as a reviewer for evaluating this enhanced manuscript. 

The authors replied in satisfactory way to my questions, while undertaking good actions to satisfy my requests and improve the quality of their work. 

Some minor editing aspects remains to be solved, but the paper can be considered as suitable for publication in Diagnostics.

Best regards,

Matteo B. Lodi